# Optimizing Time Management for Drip-and-Ship Stroke Patients Qualifying for Endovascular Therapy—A Single-Network Study

**DOI:** 10.3390/healthcare10081519

**Published:** 2022-08-12

**Authors:** Kevin Hädrich, Pawel Krukowski, Jessica Barlinn, Matthias Gawlitza, Johannes C. Gerber, Volker Puetz, Jennifer Linn, Daniel P. O. Kaiser

**Affiliations:** 1Institute and Polyclinic for Diagnostic and Interventional Neuroradiology, University Hospital Carl Gustav Carus, 01307 Dresden, Germany; 2Dresden Neurovascular Center, University Hospital Carl Gustav Carus, 01307 Dresden, Germany; 3Department of Neurology, University Hospital Carl Gustav Carus, 01307 Dresden, Germany; 4Else Kröner Fresenius Center for Digital Health, TU Dresden, 01307 Dresden, Germany

**Keywords:** stroke networks, drip and ship, endovascular therapy, quality assurance

## Abstract

BACKGROUND: We sought to identify factors for delayed drip-and-ship (DS) management in stroke patients transferred from primary hospitals to our comprehensive stroke center (CSC) for endovascular therapy (EVT). METHODS: We conducted a retrospective study of all patients transferred to our CSC for EVT between 2016 and 2020. We analyzed emergency and hospital records to assess DS process times and factors predictive of delays. We dichotomized the admission period to 2016–2017 and 2018–2020 according to the main process optimization, including the introduction of a prenotification call. RESULTS: We included 869 DS patients (median age 76 years (IQR 65–82), NIHSS 16 (IQR 11–21), 278 min (IQR 243–335) from onset to EVT); 566 were transferred in 2018–2020. Admission in 2016–2017, during on-call, longer tranfer distance, and general anesthesia were factors independently associated with delayed onset to EVT time (F(5, 352) = 14.76, *p* < 0.000). Other factors associated with delayed DS management were: transfer mode, primary hospital type, site of large-vessel occlusion, and intravenous thrombolysis. Total transfer time was faster for distances <50 km by ambulance and for distances >71 km by helicopter. CONCLUSION: Assessment of DS processes and times throughout the patient pathway allows identification of potentially modifiable factors for improvement of the very time-critical workflow for stroke patients.

## 1. Introduction

Most patients with acute ischemic stroke due to anterior circulation large-vessel occlusion (LVO) benefit from endovascular therapy (EVT) [1]. Recently, the results of the ATTENTION [2] and BAOCHE [3] trials also demonstrated the benefit of EVT in patients with basilar artery occlusion. To provide rapid EVT regardless of the geographical region, different systems of care have been developed and discussed for stroke patients. The two main organizational paradigms are (1) the mothership, in which the patient is directly brought to the EVT-capable comprehensive stroke center (CSC), and (2) the drip-and-ship (DS) model, where the initial assessment and eventual intravenous thrombolysis (IVT) at a primary hospital are followed by “shipping” to the CSC [4]. In most countries and particularly in regions outside major metropolitan areas, the density of EVT-capable CSCs is low and DS is the most common paradigm.

For both paradigms, time to EVT is crucial as it is strongly associated with functional outcomes [5]. Recently, a randomized trial found no significant difference in 90-day neurological outcomes between DS versus the mothership paradigm in patients with suspected LVO stroke [6]. However, these results were achieved on the basis of highly selected patients and optimal process times with a discrepancy of less than one hour between mothership and DS patients. Several factors influence the duration from onset to treatment. In DS patients, onset to EVT time can be mainly optimized by adjusting primary hospital and CSC processes and inter-hospital transfer.

We assessed all processes and times throughout the patient pathway from the primary hospital to our CSC to identify potentially modifiable factors for continuous improvement of the very time-critical workflow for stroke patients.

## 2. Materials and Methods

We conducted a retrospective analysis of our registry of consecutive patients with an acute ischemic stroke due to LVO who were screened for EVT at our CSC (University Hospital Dresden) between January 2016 and December 2020. We analyzed all DS patients, including baseline, emergency medical service, clinical characteristics, and process times. The study complied with the Strengthening the Reporting of Observational Studies in Epidemiology (STROBE) statement [7].

### 2.1. System of Stroke Care

Our regional stroke network is an association of 24 hospitals that provides high quality stroke expertise for the eastern part of the state of Saxony and southern part of the state of Brandenburg in Germany (Figure 1). Approximately 2.4 million people live in this area of 13,938 km^2^ [8].

The University Hospital Dresden is the main hub and coordinating CSC of the network, providing EVT and teleconsultation 24/7. The affiliated primary hospitals consist of 15 community hospitals (CH) without a dedicated neurology department/stroke unit and 8 primary stroke centers (PSC), i.e., hospitals with a neurology department and certified stroke unit. Two of them offered EVT during the study period. All hospitals in the network provided stroke imaging 24/7. A stroke fellow of the CSC supported the affiliated CHs via stroke telemedicine (telestroke) system [9].

### 2.2. Patients

We included all patients (>18 years) who were secondarily transferred to the CSC from an affiliated primary hospital. We excluded patients who were admitted directly to the CSC or suffered an in-hospital stroke, and patients who were secondarily transferred to the CSC from a hospital not affiliated to our network. We recorded the following patient baseline characteristics: age, sex, National Institute of Health Stroke Scale (NIHSS) score, location of LVO, and extent of early ischemic changes as assessed with the Alberta Stroke Program Early CT Score (ASPECTS) if the middle cerebral artery territory was involved. Good outcome was assessed with the modified Rankin scale (mRS 0–2), which indicates functional independence at 90 days after treatment.

### 2.3. Transfer and Clinical Workflow

We established standard operating procedures and quality assurance measures, including biannual meetings with interdisciplinary representatives from affiliated hospitals and the emergency medical service to optimize workflows, regular training of participating staff, morbidity and mortality conferences, and external certification during the study period.

The emergency medical service decided on the primary transport destination depending on the location where the patient presented with stroke symptoms. Generally, patients were first transported to the nearest hospital in the network. If a hospital did not have admission capacity, the CSC or another network hospital was chosen as the primary destination.

We used available clinical information and computed tomography (CT) imaging to select patients for IVT and EVT according to international recommendations [10]; CT-angiography (CTA) was mandatory to detect LVO and perfusion (CTP) imaging was optional.

For CHs, clinical and imaging findings were discussed via a telestroke platform with the stroke fellow of the CSC and patients eligible for EVT were secondarily transferred to the CSC. For PSCs, the primary indication for transfer to the CSC was made by the local neurologist. The mode of inter-hospital transfer (ambulance or helicopter) was selected by the primary hospital physicians and the emergency medical service coordinator based on transport distance and availability. At the end of 2017, we additionally offered special training to emergency medical service coordinators to increase the transport logistics of stroke patients.

The standard procedure included repeated imaging with non-contrast CT (NCCT), CTA and optional CTP at the CSC. If LVO persisted and there were no contraindication to treatment (i.e., extensive infarction or hemorrhage), we performed EVT according to international guidelines. In collaboration with the anesthesiologist, we chose the type of anesthesia for each patient (procedural sedation or general anesthesia). After treatment, patients were monitored either in a stroke unit or in a neurological intensive care unit.

Neuro-interventionalists were on-call on weekdays between 4:00 p.m. and 7:00 a.m., on weekends, and holidays. In June 2017, we implemented a team prenotification call, announcing the approximate arrival of the patient at the CSC to optimize workflow through the direct availability of stroke neurologists, neuro-interventionalists and anesthesiologists, as well as the CT scanner [11].

### 2.4. Study Objective

We aimed to analyze factors associated with delayed onset to EVT times and to identify areas of improvement. Specifically, we analyzed the following factors: admission period (dichotomized to 2016–2017 and 2018–2020 according to major quality assurance measures, including transfer optimization and prenotification call), age, gender, baseline NIHSS score on arrival at CSC, affected vascular circulation, ASPECTS, admission time (on-duty vs. on-call), type of primary hospital, IVT administration, transfer mode and distance, and type of anesthesia. We divided the onset to EVT time into (1) initial phase: symptom onset to arrival in primary hospital; (2) primary hospital phase: arrival to transfer request, including imaging and IVT admission times (if applicable); (3) transfer phase: transfer request to arrival CSC, including departure time; (4) CSC hospital phase: door of CSC to imaging and start of EVT (if applicable), Figure 2.

Secondarily, we analyzed which transfer mode (ground ambulance or helicopter) was faster depending on the distance of ground transfer (≤10 km, 11–30 km, 31–50 km, 51–70 km, 71–100 km, >100 km) between the primary hospital and the CSC. Distance categories were chosen based on the local network and infrastructure (Figure 1).

### 2.5. Statistical Analysis

Continuous variables are presented as mean and standard deviation (SD) or median and interquartile range (IQR). Categorical variables are presented as absolute and relative frequencies. We used the Shapiro–Wilk tests to test for normal distribution. We compared variables between the groups using Student’s *t*-test, Mann–Whitney U-test test or the chi-square test as appropriate. We used linear regression with stepwise selection (probability of F value for inclusion ≤ 0.050, probability of F value for exclusion ≥ 0.1) for multivariate analysis. For all statistical analyses, a *p*-value of < 0.05 was considered significant. We analyzed the data using SPSS Statistics (version 25, IBM, Armonk, NY, USA).

## 3. Results

During the study period, 1369 patients were screened for EVT at our CSC of whom 869 (63.5%) were transferred secondarily to the CSC from an affiliated primary hospital, 303 in 2016–2017 and 566 in 2018–2020, Appendix A.

### 3.1. Patient Characteristics

The median age of DS patients was 76 years (IQR 65–82), 49.3% were male, 87.3% had an anterior circulation LVO, the median NIHSS score was 16 (IQR 11–21). More than half of the patients (57.3%) received IVT and 61.4% received EVT. The most common reasons for not performing EVT were large infarct size and recanalization of the LVO before EVT. The overall rate of functional independence at 3 months was 32.4%. In 2018–2020, patients presented with a lower NIHSS score and received IVT less frequently but EVT more frequently compared with 2016–2017. The rate of helicopter transfers and general anesthesia increased in 2018–2020. Patient characteristics are shown in Table 1.

### 3.2. Factors Predicting Times of DS Patients

Among patients who received EVT, the admission period and time, transfer distance, and type of anesthesia were independently associated with the onset to EVT time, F(5352) = 14.76, *p* < 0.000, *n* = 358, adjusted(a) R^2^ = 0.162. In detail, onset to EVT was shorter for patients admitted in 2018–2020, during on-duty hours, and if the distance between the primary hospital and CSC was ≤10 km or 11–30 km. Onset to EVT was longer when patients received general anesthesia (Appendix A). To optimize the prediction of the model, we excluded 13 outliers (casewise diagnostic; outliers above three standard deviations).

We analyzed the primary hospital and transfer phases in detail because not all patients received EVT or had a documented onset time. Admission time, affected vascular circulation, IVT administration, and type of primary hospital significantly predicted primary hospital time, F(4729) = 23.28, *p* < 0.000, *n* = 734, adjusted(a) R^2^ = 0.108. Primary hospital time was shorter for patients with anterior circulation LVO, when the hospital was a PSC and if the admission was on-duty. When patients reveived no IVT, the primary hospital phase was longer (Appendix A).

Admission period, transfer distance and mode, and affected vascular circulation significantly predicted transfer time, F(8832) = 89.85, *p* < 0.000, *n* = 841, adjusted(a) R^2^ = 0.458. Transfer time was longer for longer distance transport, transport via ambulance, and patients with posterior circulation LVO. Transfer was shorter for patients admitted in 2018–2020 (Appendix A).

### 3.3. Onset to EVT Times According to Admission Period

Times of onset to EVT and of the individual phases are shown in Table 2. The median onset to EVT time decreased by 30 min when the admission periods of 2016–2017 and 2018–2020 were compared. Specifically, we found a shorter initial phase, transfer phase, and EVT hospital phase in 2018–2020.

### 3.4. Optimal Transfer Mode According to Distance

At distances ≤10 km, for ground transport, we transferred all patients by ambulance. At distances of 11–30 km and 31–50 km, the transfer time of the ambulance was faster than the helicopter transfer time, mainly due to the shorter time from requesting the transfer to departure from the primary hospital. At a distance of 51–70 km, the request time was significantly shorter but the door-to-door time from the primary hospital to CSC was significantly longer by ambulance than by helicopter, resulting in a comparable total time.

At distances of 71–100 km and >100 km, ground ambulances were more promptly available, but the transfer time was significantly longer compared with a helicopter, resulting in significantly longer times for the transfer phase by ambulance, Appendix A.

## 4. Discussion

In this study, we found that admission in 2016–2017, during on-call time, CH without stroke unit, longer transfer distance, posterior circulation occlusions, no IVT administration, and general anesthesia were associated with delays in the processes of DS stroke patients who were transferred to our CSC for EVT.

According to the admission period, the median time from onset to treatment was reduced from 300 min (IQR 255–360) in 2016–2017 to 270 min (IQR 234–318) in 2018–2020. We attribute this improvement to our quality assurance measures, including transport optimization and the introduction of a prenotification call in the CSC, because rates of admissions on-call, type of primary hospital, and affected vascular circulation were comparable in both admission periods. Interestingly, the rate of IVT administration actually declined in 2018–2020 and the rate of general anesthesia increased over the course.

The advantages of the prenotification, announcing the approximate arrival of the patient at the CSC to optimize workflow through the direct availability of stroke neurologists, neuro-interventionalists and anesthesiologists, as well as the CT scanner, for the management in the CSC have been discussed in detail in a previous study [11]. Based on the current results, we found that there is further potential for optimization, especially during on-call periods. Knowing the external image findings in advance and taking into account the clinical symptom course, a streamlined interdisciplinary decision would be possible on the degree to which repeated imaging in the CSC is necessary. One possibility to facilitate a decision on repeated imaging would be the introduction of artificial intelligence (AI)-assisted image evaluation in stroke triage for primary hospitals in the network, the results of which would also be directly available to CSC neuro-interventionalists and neurologists on-call [12,13]. Several approaches have been proposed to reduce door-to-groin time in the CSC: direct transfer to the angio suite is associated with faster treatment during all hours and treatment windows [14]. Repeated imaging may be reasonable for prolonged transfer times, significant clinical improvement or exacerbation. NCCT is often sufficient to assess the extent of infarction, rule out hemorrhage after IVT, and identify a previously seen hyperdense artery sign as an indicator of a persistent vessel occlusion [15]. If NCCT information is insufficient, additional assessment of the site of occlusion with CTA, and (if necessary) tissue at risk with CTP should be performed. Adjustment and individualization of imaging protocols for DS patients not only leads to further time savings but also to a better utilization of resources [14,16].

In patients who received general anesthesia for EVT, the delay compared with procedural sedation is not surprising because general anesthesia management, including intubation, is time-consuming [17]. In addition to the time factor, careful selection of patients eligible for sedation may also be important to achieve a better outcome [17]. However, this issue is difficult because patients with basilar artery occlusion or high NIHSS scores, in particular, often receive general anesthesia [18]. It is important that we re-evaluate our processes since the rate of general anesthesia increased in 2018–2020, although the median NIHSS score was lower and the proportion of patients with LVO in the posterior circulation remained almost the same.

Our transfer optimization resulted in shorter total transport times, i.e., from request of the transport to arrival at the CSC in 2018–2020. Interestingly, the median time from transport request to departure from the primary hospital increased slightly. Thus, the main factor for the improvement was higher travel speed. This is also consistent with the higher rate of helicopter transfers in 2018–2020. In our secondary analysis, we found that the selection of transfer mode could be further improved according to the distance between the primary hospital and CSC, i.e., for distances <51 km, ambulance and for >70 km, helicopter transfer. Our finding is in line with a previous study and highlights that the efficiency of transfer of stroke patients depends on the logistics prior to emergency medical service arrival as well as the speed of travel [19]. It is worth noting that for distances <51 km, 91.4% of patients were transported by ambulance, for 51–70 km, 36%, and >70 km, 24.5%. These numbers indicate that we had already adjusted our selections here. It remains unclear what impact the availability of the respective transport mode and the network-specific location of helicopter bases (Figure 1) had on our results.

The primary hospital phase was shorter in PSC than in CH, i.e., in hospitals without a neurology department. In CH, the attending physicians were mostly not trained neurologists, and patient assessment, imaging interpretation, as well as IVT administration and decision making towards EVT transfer, were coordinated with a stroke fellow of the CSC via telestroke service. This may indeed result in time delays. Though stroke neurologists have been found to reliably interpret the NCCT-scan of stroke patients in telemedicine [20], the introduction of AI-assisted image analysis at primary hospitals with direct availability of images and results, i.e., ASPECTS and LVO localization, to the treating physicians on site and the supporting telestroke fellow could also facilitate initial assessment and shorten the time from imaging to IVT and transfer request [12]. Interestingly, both times decreased during the study period regarding PSC and CH, which could be attributed to our quality assurance measures with training of the involved staff. However, the median time from imaging to transfer request was 51 min (IQR 35–75) in 2018–2020, which could be improved further. The discrepancy between imaging to IVT and imaging to transfer request may be due to the dilemma of frequently having the same physician responsible for administering IVT and arranging transfer during off-hours. It seems reasonable to discuss how CSC staff could further assist primary hospitals in organizing inter-hospital transfers. Recently, a study found a reduction in primary hospital door-in-door-out time in an approach whereby the ambulance crew remained with the patient on arrival. Following immediate clinical and radiological evaluation, patients were transferred to the CSC by the same ambulance crew [21]. There was no significant difference in ambulance usage time between this and the traditional approach of reordering the ambulance. The implementation of such an approach could also be discussed in our network to further reduce primary hospital and transfer times.

It is unclear why patients who did not receive IVT specifically had a longer primary hospital time. One possible explanation could be that these patients were outside the IVT time window or were admitted with unknown stroke onset time. Due to the lack of a primary treatment indication, the decision making for EVT could have been delayed. It is essential that we further investigate this phenomenon and also the delay in patients with LVO in the posterior circulation. We need to intensively train all involved staff because there is clear evidence of the benefit of EVT in certain patients in the extended or unknown time window [22,23], and more recently, in patients with basilar artery occlusion.

Finally, the time to treatment could be further shortened by using the mothership instead of the DS approach [4]. Mothership is particularly plausible in patients who have stroke symptoms in the near CSC periphery that are strongly suggestive of LVO. Several preclinical scoring systems have been developed to help emergency medical service identify these patients [4,24]. However, it should be noted that a higher number of primary admitted stroke patients with per se unclear EVT indication ties up resources in the CSC that might be needed for secondary admitted EVT-eligible patients.

The rate of functional independence at three months was 32.4% in our study, which is lower compared to the EVT groups of the randomized trials assessing the benefit of EVT in anterior circulation (46%) [1] and posterior circulation LVO (33.2%) [2]. One explanation for the lower rate is that we analyzed only those patients who were secondarily transferred for EVT and this strategy is per se associated with a longer time from onset to EVT. In addition, approximately 20% of patients in our study did not receive EVT because of large infarct size, hemorrhage, no collaterals, and no mismatch in perfusion). A detailed assessment of patient outcome and associated factors is beyond the scope of this study because there are several more factors associated with outcome (e.g., successful recanalization, treatment times, complications, etc.).

This single-network study has all the limitations of a retrospective observational design. However, we prospectively recorded the data in our institutional stroke register and the study offers homogeneity in its processes. The generalizability of our results to other networks with DS management of stroke patients remains to be evaluated.

The factors we identified may be relevant in other DS networks or in prospective studies comparing DS and mothership paradigms, and targeted evaluation and optimization should be considered. Shorter time to endovascular therapy is critical because it is strongly associated with favorable outcomes [5].

## 5. Conclusions

A detailed assessment of all processes and times throughout the patient pathway from the primary hospital to the CSC, as performed in our stroke network, allows identification of potentially modifiable factors for continuous improvement of the very time-critical workflow for stroke patients.

## Figures and Tables

**Figure 1 healthcare-10-01519-f001:**
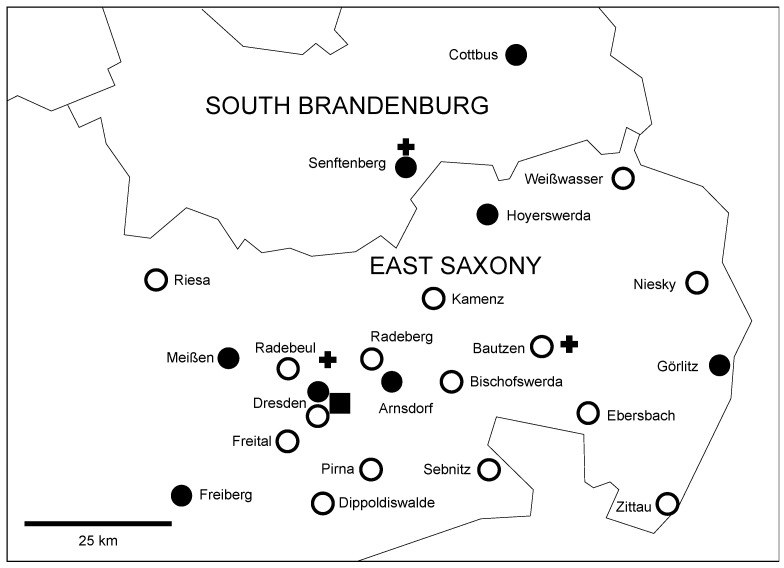
Map of the stroke network in East Saxony and South Brandenburg. Black box: comprehensive stroke center, dot: primary stroke center, ring: community hospital without stroke center, cross: helicopter base.

**Figure 2 healthcare-10-01519-f002:**
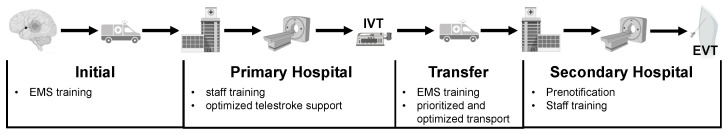
Phases of DS patient management with features of adjustment. IVT: intravenous thrombolysis, EVT: endovascular therapy, EMS: emergency medical service. Illustrations adapted from BioRender.com (2022). Retrieved from https://app.biorender.com/biorender-templates (accessed on 1 August 2022).

**Table 1 healthcare-10-01519-t001:** Patient characteristics.

	All	2016–2017	2018–2020	*p*-Value
Number of patients, *n* (%)	869	303 (34.9)	566 (65.1)	
Age, years, median (IQR)	76 (65–82)	75 (64–80)	77 (65–82)	0.201
Gender, male, *n* (%)	428 (49.3)	154 (50.8)	274 (48.4)	0.497
NIHSS, median (IQR)	16 (11–21)	17 (12–22)	16 (10–20)	0.025
Vascular circulation, *n* (%)				0.155
Anterior circulation	759 (87.3)	258 (85.1)	501 (88.5)	
Posterior circulation	110 (12.7)	45 (14.9)	65 (11.5)	
ASPECTS, median (IQR)	7 (5–8)	6 (5–8)	7 (5–8)	0.310
Primary hospital, *n* (%)				0.495
Primary stroke center	484 (55.7)	164 (54.1)	320 (56.5)	
Community hospital	385 (44.3)	139 (45.9)	246 (43.5)	
Intravenous thrombolysis, *n* (%)	498 (57.3)	205 (67.7)	293 (51.8)	0.000
Transfer distance, *n* (%)				0.271
≤10 km	17 (2)	8 (2.6)	9 (1.6)	
11–30 km	283 (32.6)	100 (33)	183 (32.3)	
31–50 km	130 (15)	51 (16.8)	79 (14)	
51–70 km	150 (17.3)	44 (14.5)	106 (18.7)	
71–100 km	94 (10.8)	27 (8.9)	67 (11.8)	
>100 km	195 (22.4)	73 (24.1)	122 (21.6)	
Transfer mode, *n* (%)				0.012
Ground ambulance	518 (59.6)	198 (65.3)	320 (56.5)	
Helicopter	351 (40.4)	105 (34.7)	246 (43.5)	
On-call admission	531 (61.1)	189 (62.4)	342 (60.4)	0.574
Endovascular therapy, *n* (%)	534 (61.4)	163 (53.8)	371 (65.5)	0.000
Reason for no EVT, *n* (%)				
Large infarct size	157 (18.1)	67 (22.1)	90 (15.9)	
Hemorrhage	1 (0.1)	0	1 (0.2)	
No collaterals	1 (0.1)	1 (0.3)	0	
LVO recanalization	139 (16)	56 (18.5)	83 (14.7)	
No mismatch in perfusion	10 (1.2)	4 (1.3)	6 (1.1)	
other life limiting conditions	2 (0.2)	2 (0.7)	0	
randomization	1 (0.1)	0	1 (0.2)	
others	23 (2.6)	10 (3.3)	12 (2.3)	
General anesthesia, *n* (%)	383 (71.7)	101 (62)	282 (76)	0.001
mRS 0–2 at 90 days, *n* (%)	249 (32.4)	88 (29.6)	161 (34.2)	0.189

IQR: interquartile range, NIHSS: National Institutes of Health Stroke Scale, ASPECTS: Alberta stroke program early CT score, EVT: endovascular therapy, LVO: large vessel occlusion, mRS: modified Rankin scale.

**Table 2 healthcare-10-01519-t002:** Times of drip-and-ship patients.

	All Median (IQR)	2016–2017 Median (IQR)	2018–2020 Median (IQR)	*p*-Value
Onset to EVT, min (*n* = 374)	278 (243–335)	300 (255–360)	270 (234–318)	0.001
Onset to IVT, min (*n* = 439)	113 (88–145)	115 (90–150)	110 (87–141)	0.259
Initial phase, min (*n* = 551)	63 (45–92)	67 (48–100)	62 (43–88)	0.037
Primary hospital phase, min (*n* = 748)	72 (53–101)	75 (56–104)	71 (51–99)	0.115
Arrival–Imaging (*n* = 761)	18 (11–27)	17 (10–27)	18 (12–28)	0.091
Imaging–IVT (*n* = 490)	25 (13–37)	27 (16–40)	22 (12–35)	0.008
Imaging–Transfer request (*n* = 844)	53 (36–77)	56 (39–81)	51 (35–75)	0.030
Transfer phase, min (*n* = 857)	71 (58–89)	73 (59–94)	69 (58–86)	0.043
Request–Departure (*n* = 857)	33 (25–48)	32 (24–48)	34 (26–48)	0.131
Departure–Door CSC (*n* = 858)	34 (27–43)	37 (28–48)	33 (27–41)	0.001
CSC phase, min (*n* = 531)	63 (51–76)	69 (56–84)	62 (51–73)	0.000
Arrival– Imaging (*n* = 855)	13 (9–18)	15 (10–21)	12 (9–16)	0.000
Imaging–EVT (*n* = 522)	50 (39–62)	53 (41–66)	49 (38–61)	0.019

IQR: interquartile range, EVT: endovascular therapy, IVT: intravenous thrombolysis, CSC: comprehensive stroke center.

## Data Availability

The data presented in this study are available on request from the corresponding author.

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
