# Peer review of "Optimizing Time Management for Drip-and-Ship Stroke Patients Qualifying for Endovascular Therapy—A Single-Network Study"

_healthcare, 2022, doi:10.3390/healthcare10081519_

Round 1

Reviewer 1 Report

In this study, the auhors aimed  to identify factors for delayed drip-and-ship (DS) manage ment in stroke patients.

This is a retrospective observational study. The importance of observational studies lies in the identification of important disorders and providing impetus to future research[1]

They only assessed times and not the impact of drip-and-ship (DS) manage ment on patient’s outcome.

What does this study add a new data to current medical knowledge?

References

1.        Polat HB, Kanat A, Celiker FB, et al (2020) Rationalization of using the MR diffusion imaging in B12 deficiency. Ann Indian Acad Neurol 23:72–77. https://doi.org/10.4103/aian.AIAN_485_18

Author Response

Response to Editor and Reviewers

Dear Editors,
Dear Reviewers,

We would like to resubmit a revised version of our manuscript “Optimizing time management for drip-and-ship stroke patients qualifying for endovascular therapy - a single-network study.”

We thank the reviewers for their constructive and helpful suggestions. We provide our response to the reviewers’ comments below and believe that the quality of our manuscript has improved as a result.

Reviewer 1 Comment to the Author:

In this study, the authors aimed to identify factors for delayed drip-and-ship (DS) management in stroke patients.

This is a retrospective observational study. The importance of observational studies lies in the identification of important disorders and providing impetus to future research [1]

References

  1. Polat HB, Kanat A, Celiker FB, et al (2020) Rationalization of using the MR diffusion imaging in B12 deficiency. Ann Indian Acad Neurol 23:72–77. https://doi.org/10.4103/aian.AIAN_485_18

Thank you for the brief summary.

They only assessed times and not the impact of drip-and-ship (DS) management on patient’s outcome.

Thank you for your feedback. We focused on the management before endovascular therapy. A detailed assessment of patient outcome is beyond the scope of this study because there are several more factors associated with outcome (e.g., successful recanalization, treatment times, complications, etc.).

What does this study add a new data to current medical knowledge?

In our single-network study, we found that several factors (e.g., transfer strategy based on transfer distance) were associated with delayed initiation of treatment in patients transferred by drip and ship. Our study shows that a detailed assessment of DS processes and times across the patient pathway allows identification of potentially modifiable factors to improve the highly time-critical workflow of stroke patients. The factors we identified may be relevant in other DS networks or in prospective studies comparing DS and mothership paradigms, and targeted evaluation and optimization should be considered. Shorter time to endovascular therapy is critical because it is strongly associated with favorable outcome.

We have added a paragraph to the Discussion (please also see the response to the second-last comment from reviewer 2).

Reviewer 2 Comment to the Author:

The manuscript is well-written and addresses an important topic on optimizing time management for drip-and-ship stroke patients qualifying for Endovascular therapy

Thank you.

Suggestion to add the abbreviations in the manuscript.

In the "Instructions for Authors" and in the template for the article we did not find if and where we can put an additional list of abbreviations. We would like to include the following list in the manuscript and thank the editorial team in advance for their help.

AI                    artificial intelligence

ASPECTS     Alberta Stroke Program Early CT

CH                  community hospital

CSC               comprehensive stroke center

CTA                computed tomography angiography

CTP                computed tomography perfusion

DS                  drip-and-ship

EVT                endovascular therapy

IQR                 interquartile range

IVT                  intravenous thrombolysis

LVO                large-vessel occlusion

mRS               modified Rankin scale

NCCT             non-contrast CT

NIHSS           National Institute of Health Stroke Scale

PSC               primary stroke center

SD                  standard deviation

Abstract: pls add median for the age

We have added “median age” in the Abstract (page 1 line 18).

There were repetitions in the objectives (page 2, line 49) and page 3, line 119.

We have changed the last paragraph of the introduction (page 2 line 49-51):

“We assessed all processes and times throughout the patient pathway from the primary hospital to our CSC to identify potentially modifiable factors for continuous improvement of the very time-critical workflow for stroke patients.”

It reported that the overall rate of functional independence at three months was 32.4%. Please discuss this finding and compare it with other studies.

The rate of functional independence at three months was 32.4% in our study, which is lower compared to the EVT groups of the randomized trials assessing the benefit of EVT in anterior circulation (46%) [1] and posterior circulation LVO (33.2 %) [2]. One explanation for the lower rate is that we analyzed only those patients who were secondarily transferred for EVT and this strategy is per se associated with a longer time from onset to EVT. In addition, approximately 20% of patients in our study did not receive EVT because of large infarct size, hemorrhage, no collaterals, and no mismatch in perfusion). A detailed assessment of patient outcome and associated factors is beyond the scope of this study because there are several more factors associated with outcome (e.g., successful recanalization, treatment times, complications, etc.).

We have added this paragraph to the Discussion (page 8 line 297-306).

  1. Goyal, M.; Menon, B.K.; Van Zwam, W.H.; Dippel, D.W.J.; Mitchell, P.J.; Demchuk, A.M.; Dávalos, A.; Majoie, C.B.L.M.; Van Der Lugt, A.; De Miquel, M.A.; et al. Endovascular Thrombectomy after Large-Vessel Ischaemic Stroke: A Meta-Analysis of Individual Patient Data from Five Randomised Trials. Lancet 2016, 387, 1723–1731, doi:10.1016/S0140-6736(16)00163-X.
  2. Tao, C.; Li, R.; Zhu, Y.; Qun, S.; Xu, P.; Wang, L.; Zhang, C.; Liu, T.; Song, J.; Sun, W.; et al. Endovascular Treatment for Acute Basilar Artery Occlusion: A Multicenter Randomized Controlled Trial (ATTENTION). International Journal of Stroke 2022, 174749302210771, doi:10.1177/17474930221077164.

Suggest having a separate paragraph for the recommendation based on the findings of this study and potential study or investigation in the future.

We have added the following paragraph to the Discussion (page 8 line 311-314):

“The factors we identified may be relevant in other DS networks or in prospective studies comparing DS and mothership paradigms, and targeted evaluation and optimization should be considered. Shorter time to endovascular therapy is critical because it is strongly associated with favorable outcome.”

The conclusion should reflect the answer of the objectives

We have changed the objective (see above) and the conclusion now reflects the answer of the objectives.

Again, we thank the reviewers for their helpful comments and the chance to improve the quality of our manuscript. We are our looking forward to your response with regards to our revisions.

Sincerely,

Daniel Kaiser

Reviewer 2 Report

The manuscript is well-written and addresses an important topic on optimizing time management for drip-and-ship stroke patients qualifying for Endovascular therapy

The manuscript is well-written and addresses an important topic on optimizing time management for drip-and-ship stroke patients qualifying for endovascular therapy.

Suggestion to add the abbreviations in the manuscript.

Abstract: pls add median for the age

There were repetitions in the objectives (page 2, line 49) and page 3, line 119. 

It reported that the overall rate of functional independence at three months was 32.4%. Please discuss this finding and compare it with other studies.

Suggest having a separate paragraph for the recommendation based on the findings of this study and potential study or investigation in the future.

The conclusion should reflect the answer of the objectives

Author Response

(The authors gave the same response as above.)
